# Improvement and Optimization of Electromagnetic Integrated Scanning Micromirror

**DOI:** 10.3390/mi12101213

**Published:** 2021-10-04

**Authors:** Liang Chen, Wenwen Gu

**Affiliations:** College of Engineering and Technology, Southwest University, Chongqing 400715, China; top626116@email.swu.edu.cn

**Keywords:** driving system, integrated scanning micromirror, driving voltage, mechanical scanning angle

## Abstract

In this paper, the effect of driving system on working performance of electromagnetic integrated scanning micromirror (ISM) is studied. To further improve the optimization design of the electromagnetic ISM, the detailed theoretical analysis, simulation analysis, and experimental test are carried out, respectively. By changing the force form and external magnetic field of the device, the mechanical scanning angle, driving voltage, and resonant frequency of the electromagnetic ISM will be changed accordingly, and then the change of the working performance of the ISM is explored. Through the optimization analysis and comparative test, the optimal design scheme of driving system is obtained, and the effect of driving system on the working performance of the electromagnetic ISM is verified. The experimental results show that by optimizing the structure of the driving system, the mechanical scanning angle of the electromagnetic ISM is increased by about 20%, and the driving voltage is decreased about 10% observably, and the working performance of the electromagnetic ISM is significantly improved. The research results have important significance and practical application value for the extended application of the electromagnetic ISM in the field of optical detection.

## 1. Introduction

Electromagnetic ISM is an optical chip which integrates the optical reflector and micro driver using MEMS technique [1,2]. Compared with the traditional optical scanning devices, the electromagnetic ISM has the advantages of small size, low power consumption, fast response, etc. In recent years, with the rapid development of the MEMS manufacturing technique, the electromagnetic ISM has been widely used in consumer electronics [3,4], food safety [5,6,7], medical research [8], and other fields [9,10,11,12].

The electromagnetic ISM utilizes electromagnetic driving principles to realize mechanical work, which is mainly composed of the driving system structure and ISM entity structure. The structure of the driving system of the electromagnetic ISM is composed of driving signal, driving coil, and magnetic field, and the ISM entity structure mainly includes integrated circuit and frame structure. It is known that changing the driving signal will change the natural frequency of the ISM [13,14], which will affect the mechanical scanning performance of the ISM. Therefore, in order to improve the effect of the driving system on the working performance of the electromagnetic ISM, this paper carries out the corresponding research and analysis of the driving coil and magnetic field on the working performance of the electromagnetic ISM, aiming to provide supplementary research for the design and optimization of the electromagnetic ISM. 

## 2. Structure and Working Principle

The main structure of the electromagnetic ISM is shown in Figure 1. It is mainly composed of fixed frame, micromirror, torsion beams, lead wires, driving coil, permanent magnetics, electrodes, etc. The lead wires are integrated on the torsion beams. The two torsion beams are symmetrically distributed about the center of the micromirror to connect the fixed frame and the micromirror. The driving coil is integrated on the back of the micromirror and expanded from the inside to the outside in an involute way. The open ends of the driving coil are connected to the electrodes through the lead wires. In order to realize electromagnetic drive, the two permanent magnets are placed symmetrically on both sides of the ISM, which are used to provide a constant permanent magnetic field. Therefore, the whole structure of the electromagnetic ISM is completed. The driving coil, permanent magnets, and external power supply device together constitute the driving system of the electromagnetic ISM.

Under the action of the magnetic field, when an alternating current is fed into the driving coil, the straight lines of the driving coil in parallel with the torsion beams will generate Lorentz force. As the alternating current in the driving coil distributed on both sides of the micromirror flows in opposite directions, the Lorentz forces generated on both sides of the micromirror are also opposite. Therefore, the Lorentz forces will drive the micromirror to rotate back and forth with the change of alternating current. When the driving frequency applied into the driving coil is consistent with the natural frequency of the ISM, the electromagnetic ISM will work in resonance state and the mechanical scanning angle will reach the maximum value [15]. 

## 3. Simulation Analysis and Design

The structure of the driving system and the ISM are integrated together using silicon material. According to the characteristics of the electromagnetic ISM, if the structure of the driving system changes, the working performance of the ISM may be affected to some extent. On this basis, the professional software ANSYS and Maxwell are used for simulation to research the effect of the driving coil and magnetic field on the working performance of the electromagnetic ISM. 

As shown in Figure 2, in order to study the effect of the driving coil on the working performance of the electromagnetic ISM, the six surfaces, labeled A, B, C, D, E, and F, respectively, with the same width of 0.02 mm, are taken as the acting surfaces of Lorentz force. Surfaces A, B, C, and surfaces D, E, F are symmetrically distributed with equal dimensions about the central axis of the torsion beams. When current *i* is applied to the driving coil, Lorentz force generated on each surface can be expressed as:(1)$$F_{n}=B_{n}l_{n}i$$
where, *l_n_* represents the *n*-th length of the six corresponding surfaces, and *B_n_* represents the magnetic field intensity at the location of the nth surface. 

To study the distribution characteristics of the magnetic field generated by the two permanent magnets, the professional magnetic field simulation software Maxwell is used to carry out the magnetic field simulation. In the settings, a two-dimensional plane model is used to carry out the simulation. The value of the remanent magnetic field is set to 1.4 T, and the coercivity force is 890 kA /m. The size of the bar magnet is 20 mm × 3 mm and the magnet spacing is 12 mm. The center axis of the torsion beams is set to the Y-axis and the center of the micromirror is set to the origin coordinates. Through simulation analysis, by taking data points at equal intervals along the X-axis of the simulation results and taking the highest fifth power of the variable, the equation of the magnetic field intensity changing with the position can be expressed as:(2)$$\begin{array}{ll} B_{x} & =-1.979x^{5}-0.005x^{4}+3.295x^{3}\\ & +\;0.341x^{2}-9.237x+95.653 \end{array}$$

From the fitting formula of Equation (2), the intensity of the magnetic field is the weakest at the center of the micromirror and increases symmetrically as it approaches the permanent magnets on both sides. Moreover, the trend of growth is curved and uneven. That is to say, the ISM does not work in a uniformly distributed magnetic field. The magnetic field at the position of the longest driving coil in the outermost turn is the strongest, which can provide the maximum Lorentz force for the mechanical working of the electromagnetic ISM under the action of the magnetic field.

Therefore, to study the effect of the driving system on the working performance of the electromagnetic ISM, the statics simulation analysis is carried out. In the specific simulation setting, the grid of the torsion beams is refined to improve the reliability of the simulation results, and the grid element size is set to 0.06 mm. The size of the micromirror is 6 mm × 7 mm. The vector length of the torsion beam is 3.6 mm, and the width of the torsion beam is 0.15 mm. In addition, the fixed frame, torsion beam, and micromirror are made of monocrystalline silicon with the same thickness of 0.3 mm. 

According to Equation (2), the magnetic field intensity is not uniformly distributed in the micromirror area. It is known that the driving coil is designed in involute mode [16], and the length of each straight line increases from inside to outside. Therefore, under the action of different magnetic field intensity, the force generated by each straight driving coil is also different [17,18]. In the specific simulation, the force calculated by the Equations (1) and (2) is applied on the both sides of the micromirror, and the mechanical performance of the ISM is obtained and analyzed by recording the stress value and displacement generated by the simulation. In order to provide sufficient simulation data, the forms of applying force are divided into two forms: bilateral and unilateral, and the simulation settings and results are shown in Table 1. 

As exhibited in Table 1, it can be seen from the results of sequence numbers 1–4 that when the symmetrical forces exerted on the micromirror are multiplied or changed irregularly, the ratio of maximum stress to maximum displacement is almost unchanged, which has been about 683.35. In contrast, from the simulation results of the unilateral force form of the micromirror with sequence numbers 5–8, when the three forces on one side are removed, the ratio changes significantly. The simulation results show that when the stress condition changes, the working performance of the electromagnetic ISM may also change with, and the greater the force applied, the greater the ratio. Based on the simulation results exhibited above in Table 1, in order to study the effect of driving system on the working performance of the electromagnetic ISM, some experiments will be carried out to demonstrate the research. 

From Equations (1)–(2), it can be seen that the optimization of driving coil and magnetic field is to obtain the most reasonable force for the electromagnetic ISM, and the two physical quantities have the same property on the formation of the force. Therefore, to simplify the research scheme, only the influence of magnetic field on the working performance of the electromagnetic ISM be carried out in this paper. Three different permanent magnets, N35, N38, and N52, are designed and fabricated, which are used to combine with ISMs to carry out experimental tests. The property and spacing settings of the permanent magnets are shown in Table 2. 

## 4. Test and Analysis

Figure 3a shows the physical photo of the electromagnetic ISM, which mainly consists of two permanent magnets, interfaces, cover plate, PCB board (printed circuit board), and baseplate. The cover plate and baseplate are used for fixing the middle layer ISM chip and PCB board. In the middle layer, the ISM chip is adhered to the center of the PCB. Then, two slots for placing permanent magnets are designed on both sides of the ISM chip on the PCB. Therefore, the two permanent magnets are fixed by the cover plate and baseplate layers. Besides, at the lower end of the PCB board, there are several circuit interfaces for connecting external driving signals. These interfaces are electrically connected to the ISM through gold wires inside the PCB board. Figure 3b,c show the chip diagram of the electromagnetic ISM. The ISM is fabricated on silicon substrate by silicon micromachining techniques, and the thickness of the micromirror is about 0.3 mm. The blazed grating with a blazed angle of 7.9° is fabricated by etching technology on the front of the micromirror, which is used for optical work. On the back of the micromirror is a layer of driving coil fabricated by dry etching technology. Under the action of the permanent magnets, when an alternating current is applied to the driving coil, the coil will generate Lorentz force to drive the micromirror to move mechanically. In addition, an angle sensing coil is integrated on the outer side of the driving coil (the four angle plane areas on the back of the micromirror), which generates electromotive force by cutting the magnetic induction lines to realize real-time monitoring of the deflection state of the micromirror. At last, through inductively coupled plasma etching [19], a pair of torsion beams and the micromirror are released.

Figure 4 shows the test platform based on the working principle of the electromagnetic ISM. The test instrument mainly consists of a position sensitive detector (DRX-1DPSD-OA03-X, PSD) and a red spot laser (650 nm). In the non-working state, the red spot emitted by the red spot laser will be projected onto the front of the micromirror, and then reflected to the middle position of the PSD. When an alternating current is applied to the driving coil, the red spot projected on the PSD will form a continuous straight scanning line due to the high-speed deflection of the micromirror. The red dot on the micromirror and the two ends of the straight scanning line on the PSD will form an equilateral triangle. Therefore, the mechanical scanning angle *θ* of the electromagnetic ISM can be obtained by using Pythagorean theorem.

In order to provide sufficient test data, two kinds of ISM chips with straight torsion beam and folded torsion beam are designed and fabricated, respectively. During the test, permanent magnets N35 and N38 are combined with the ISM chip with straight torsion beams for the experimental test, and the ISM chip with folded torsion beam will be tested in combination with the permanent magnets N38 and N52, respectively. The test results of the electromagnetic ISM with straight torsion beam are shown in Table 3, and the test results of electromagnetic ISM with folded torsion beam are shown in Table 4. 

Table 3 shows the test results of four electromagnetic ISMs with straight torsion beam. Under the action of the permanent magnet N35, the maximum mechanical deflection angle produced by the electromagnetic ISM is 2.11° and 2.24°, and the driving voltage are 5.12 Vpp (peak-to-peak voltage) and 7.50 Vpp, respectively. In contrast, under the action of permanent magnet N38 with stronger magnetism, the maximum mechanical deflection angle of the electromagnetic ISM can reach 4.89° and 4.81°, and the driving voltage are only 2.72 Vpp and 2.84 Vpp, respectively. The test results show that by optimizing the design of magnetic field, the electromagnetic ISM can obtain larger mechanical scanning angle and lower driving voltage. Not only can the mechanical scanning angle of the electromagnetic ISM be effectively improved, but the driving voltage can be significantly decreased. 

Table 4 shows the comparative test results of two electromagnetic ISM chips with folded torsion beam. It can be seen that under the action of permanent magnet N38 with weaker magnetism, the maximum mechanical deflection angle of the ISM B1 and ISM B2 are 5.85° and 6.03°, respectively. When the two ISMs are combined with the permanent magnet N52, the maximum mechanical deflection angle of the two electromagnetic ISMs can reach 7.03° and 7.08°, respectively. The maximum deflection angle increased by 20.2% and 17.4%. The driving voltage dropped from 1.51 Vpp and 2.57 Vpp to 1.41 Vpp and 2.35 Vpp, a decrease of 6.6% and 8.6%, respectively. Furthermore, the maximum deflection angle of the ISM with folded torsion beam can reach 7.08°. Theoretically, the full scanning angle of the ISM can reach 14.16° (the full scanning angle is twice the mechanical deflection angle). Moreover, the lowest driving voltage is only 1.41 Vpp. Generally, many other electromagnetic ISMs, even the ISMs with other driving modes, need a voltage of up to tens of volts to drive, but can only obtain a scanning angle of less than 10° [20,21,22]. Therefore, the research results of this paper can advantageously facilitate the development of the ISM in large scanning angle and low voltage. In addition, it can be seen that the resonance frequency of the two electromagnetic ISMs has shifted, indicating that the change of the driving system can not only change the performance of the electromagnetic ISM, but also affect the structural characteristics of the ISM. 

Based on the comparative test results, it is verified that the working performance of the electromagnetic ISM is affected by the driving conditions. When the driving conditions change, the structural performance of the electromagnetic ISM will also change. By optimizing the driving system, not only can the deflection angle of the electromagnetic ISM be improved prominently, but the driving voltage can be effectively reduced distinctly, so that the electromagnetic ISM can work under a lower driving voltage. This will be of great significance to the development of large scanning angle and low voltage drive of the electromagnetic ISM [23,24].

## 5. Conclusions

To study the effect of the driving system on the working performance of the electromagnetic ISM, theoretical simulation and an experimental test are carried out and analyzed. Through the theoretical simulation analysis, the main parameters affecting the ISM working performance are determined, including structure size, magnetic field, and driving signal. When the magnetic field interacts with the driving signal to produce Lorentz force, it will affect the working performance of the electromagnetic ISM. Then, two ISM chips with different structures and three permanent magnets with different magnetic properties are designed and fabricated for test. The tests results show that the mechanical scanning angle of the electromagnetic ISM is increased by about 20%, and the driving voltage is decreased about 10%, which makes the electromagnetic ISM obtain larger scanning angle and work under lower driving voltage. It is concluded that by optimizing the driving system, not only can the scanning angle of the electromagnetic ISM be improved, but the driving voltage can be effectively reduced distinctly. The working performance of the electromagnetic ISM is significantly improved. The research results are of great significance for the further development of the electromagnetic ISM in the optical field.

## Figures and Tables

**Figure 1 micromachines-12-01213-f001:**
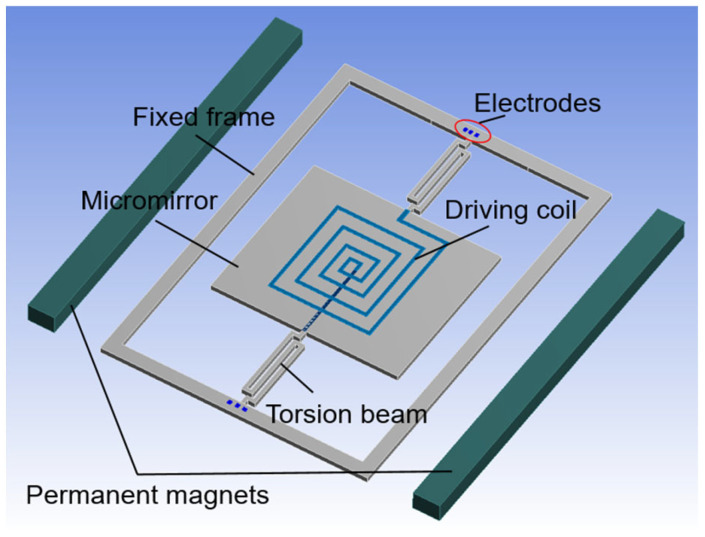
The schematic diagram of the electromagnetic ISM.

**Figure 2 micromachines-12-01213-f002:**
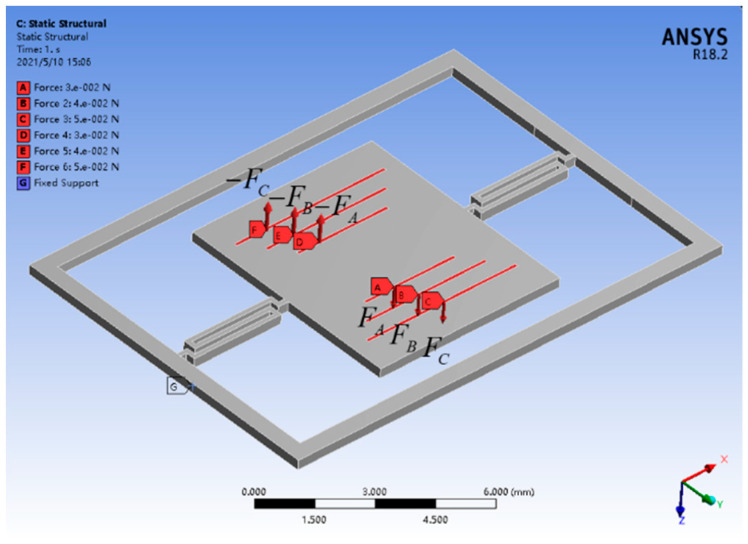
Simulation setting diagram of the ISM.

**Figure 3 micromachines-12-01213-f003:**
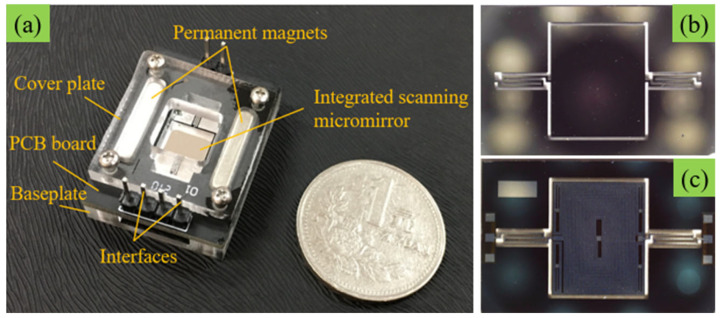
(**a**) The photograph of the electromagnetic ISM, (**b**) front side of the ISM chip, (**c**) back side of the ISM chip.

**Figure 4 micromachines-12-01213-f004:**
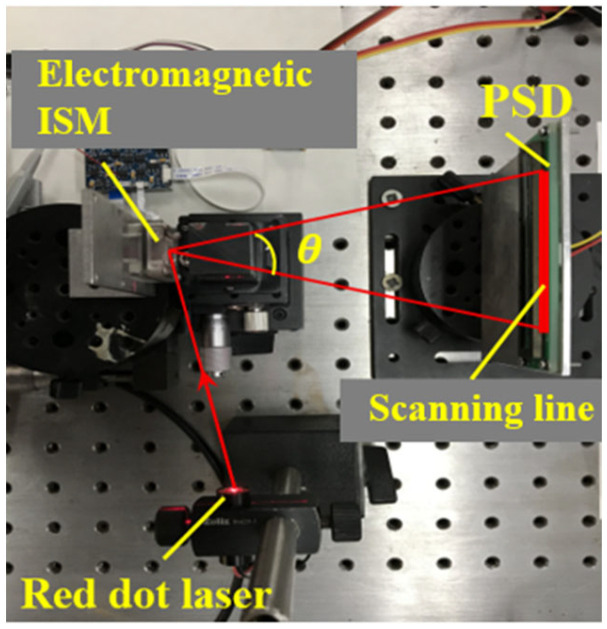
Test setup of the electromagnetic ISM.

**Table 1 micromachines-12-01213-t001:** Simulation results of driving system on working performance of the electromagnetic ISM.

Number	Type	Fore [*F*_A_, *F*_B_, *F*_C_] (N)	Ratio of Maximum Stress to Maximum Displacement
1	Bilateral force	0.010, 0.012, 0.014	683.77
2		0.020, 0.024, 0.028	683.81
3		0.024, 0.028, 0.032	683.81
4		0.032, 0.034, 0.036	683.81
5	Unilateral force	0.010, 0.012, 0.014	708.39
6		0.020, 0.024, 0.028	710.15
7		0.024, 0.028, 0.032	708.75
8		0.032, 0.034, 0.036	709.88

**Table 2 micromachines-12-01213-t002:** Property and spacing of the permanent magnets N35, N38, and N52.

Mark	Remanence (T)	Coercive Force (KA/m)	Size (mm^3^)	Spacing (mm)
N35	1.22	890	15 × 3 × 5	22
N38	1.25	910	15 × 3 × 6	12
N52	1.39	915	20 × 3 × 10	10.6

**Table 3 micromachines-12-01213-t003:** Test results of electromagnetic ISM with straight torsion beam.

Mark	ISM Chip	Driving Coil Resistance	Driving Voltage	Maximum Deflection Angle
N35	A1	56.05 Ω	5.12 Vpp	2.11°
	A2	59.58 Ω	7.50 Vpp	2.24°
N38	A3	69.49 Ω	2.72 Vpp	4.89°
	A4	66.10 Ω	2.84 Vpp	4.81°

**Table 4 micromachines-12-01213-t004:** Comparative test results of electromagnetic ISM with folded torsion beam.

Mark	ISM Chip	Driving Coil Resistance	Resonant Frequency	Driving Voltage	Maximum Deflection Angle
N38	B1	90.13 Ω	520.0 Hz	1.51 Vpp	5.85°
	B2	113.13 Ω	604.5 Hz	2.57 Vpp	6.03°
N52	B1	90.13 Ω	523.8 Hz	1.41 Vpp	7.03°
	B2	113.13 Ω	604.4 Hz	2.35 Vpp	7.08°

## Data Availability

Not applicable.

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
