# Peer review of "Improvement and Optimization of Electromagnetic Integrated Scanning Micromirror"

_micromachines, 2021, doi:10.3390/mi12101213_

Round 1

Reviewer 1 Report

  1. The authors should add more literature review with focus on methodology.
  2. The abstract talk about the optimization, the authors should talk more about that factor in methods section
  3. The results are difficult to follow, the authors should add figures perhaps
  4. the conclusion is not so clear, it should be simplified 

Author Response

According to the review comments, the revised replies are as follows:

  1. As suggested, we have added technical research references to the paper.
  2. According to the revised opinions, we have completed the optimization method in the abstract part.
  3. In order to highlight the innovation and make it easier for readers to understand the paper, we have added some necessary data in paragraph 4 of section 4 of the paper.
  4. According to the review comments, we have reorganized the conclusions and simplified it appropriately.

Reviewer 2 Report

The authors report a technology of interest for an EM integrated scanning laser mirror. The technology of the device is correctly introduced but several questions merit to be considered for the final form of the paper : 

Develop with more details the comparaison of the performances and characteristics of the MEMS and the ISM technologies for laser beam scanning.

  • Also include performances of the ISM in terms of speed, spatial resolution.
  • Does the stress on the mirror induces a spatial deformation of its surface.   
  • Typical values of the current in the driving coils.
  • It  is shown on table 4 that a high deflection angle of 7.03° is obtained with a low driving voltage. Does it result from an optimization of the structure and of the operating conditions. Is it a correct agreement between the theoretical modeling and the experimental results.

 To conclude the manuscript desserves to be improved and completed by answering to questions and comments before being accepted for publication in the journal. 

Author Response

According to the review comments, the revised replies are as follows:

  1. According to the review comments, we have supplemented the necessary data in Table 4 and given explanations in the corresponding paragraphs.
  2. We have supplemented the necessary data in Table 4 and given explanations in the corresponding paragraphs.
  3. According to figure 1, the micromirror is suspended in space through two torsion beams. When the micro mirror is mechanically deflected, the stress mainly occurs at the connection between the beam and the micro mirror. At the end of each test, we will carry out microscopic inspection on the external surface of the device. After inspection, it is found that the main stress areas of all devices are intact without deformation or warping.
  4. In this paper, two ISM with straight beam and folded beam are used. The typical currents applied into the driving coil are about 15 mArms and 8 mArms respectively.
  5. This kind of ISM with a deflection angle of 7.03°is a new structure developed by our research group in recent years. The ISM adopts the folded beam structure as the torsion beam, which has better mechanical torsion performance than the straight beam. Through the structural optimization design and simulation analysis, the structure has better mechanical deflection ability than the straight beam structure, and the theory is also proved by our comparative experiments.
  6. According to the review opinions, we have repeatedly revised and combed the full text. For the existing problems, we have made necessary supplementary explanations and made corresponding corrections.

Reviewer 3 Report

In review of the paper titled "Research on the effect of driving system on working characteristics of electromagnetic integrated scanning micromirror," their are significant changes that should be made.

  1. There is no fabrication approach or methods outlined in the paper so one has to question whether this device was physically developed and if so, how?
  2. Few dimensions are provided on the physical design and why those dimensions were selected. There is no supporting or comparative documentation to justify why the physical dimensions were selected.
  3. What was the ultimate goal of the research? For example, what was the scanning angle you were trying to achieve, what is the resonance of the device or scanning speed you were trying to achieve?
  4. How does your design even enable scanning? What are the gaps to enable scanning? Again, how was this device fabricated.
  5. What was your method or approach to the positioning of the permanent magnets? Distance away from micromirror structure, magnet thickness choices, how did you place the magnets, by hand or some other method?
  6. What is the stress and strain levels within the structure during scanning? Will have a major effect on reliability of the device. 
  7. Document needs to include images of the fabricated device, not just a simulated design, SEM or optical image is best, images of the simulated results to help support the various tables presented, etc. 
  8. The two images provided need to be enlarged as one cannot easily read the values. 
  9. Why is your scanning angle so small, many researchers have other designs that exhibit much greater scanning angles than the one presented for mirrors much smaller than the one presented. 
  10. I feel you could significantly expand on the device characteristics, fabrication, and testing to help support the overall testing in addition to supporting your conclusions 
  11. Not sure how valuable the research is as there are a lot of researchers developing scanning micromirrors with much larger scan angles. You should expand your references to include some of these prior works and state what you are doing and how it is unique or better than what is currently being fielded. 

Author Response

According to the review comments, the revised replies are as follows:

  1. This paper mainly focuses on the research on the driving system of the electromagnetic ISM, so the manufacturing is not introduced in detail. According to the comments, and in order to facilitate readers' understanding, we have quoted the article [19] on ISM fabrication approach in paragraph 2 of section 4.
  2. This paper mainly focuses on the research on the driving system of the electromagnetic ISM, so the source of the dimensions is not introduced in detail. In order to improve the content of the paper, we have quoted the article [18] on ISM dimensional optimization design in paragraph 2 of section 4.
  3. The goal we want to achieve is: through comparative test, the optimized ISM can obtain larger scanning angle and lower driving voltage. This is to promote the development of ISM in large scanning angle and low voltage driving. Through theoretical analysis and experimental test, it is found that the working index of the device can be effectively improved by optimizing the driving system.
  4. â‘ The working principle of ISM is introduced in paragraph 2 of section 2. When the driving current is applied to the equipment, the micromirror will be mechanically deflected. Then, Fig. 3 introduces the test schematic diagram of the ISM. When the light source (red dot laser) is projected to the ISM, the mechanical scanning can be realized under the rotation of the micro mirror.â‘¡ As shown in Figure 1, the micromirror is suspended in the fixed frame. In order to avoid other structures affecting the deflection of the micromirror, appropriate gaps are set around the micromirror and at the entrance of the light source through optimal design.â‘¢In order to facilitate readers' understanding, we have quoted the article [19] on ISM fabrication approach in paragraph 2 of section 4.
  5. â‘  The package of electromagnetic ISM is divided into three layers. The upper and lower layers are mainly used for fixing the middle layer ISM chip. In the middle layer, the ISM chip is adhered to the center of the PCB. Then, according to the position design, slots for placing permanent magnets are designed on both sides of the ISM chip on the PCB. The permanent magnets are fixed by the upper and lower layers. â‘¡ The distance and size are obtained through the optimal simulation of the magnetic field, and the specific data have been given in Table 2.â‘¢ In order to ensure the packaging of the device, the magnets are fixed with experimental apparatus in the dust-free laboratory.
  6. Through the static analysis of the ISM, when the ISM reachs the maximum mechanical scanning angle, the maximum stress generated in the device is about 0.5 GPa, which is far lower than the allowable stress of silicon material of 1.5 GPa, which will not affect the structural reliability of the device. Moreover, after each test, the ISM chip will be visually inspected. The inspection results show that the surface of the ISM is intact without crack or warpage, and the device can work normally after power on.
  7. According to the review comments, as shown in Figure 3, we have added the physical picture and chip photo of electromagnetic ISM in section 4, and given explanations in the corresponding paragraph.
  8. According to the review suggestions, the pictures in the paper have been properly enlarged.
  9. Since the device is driven by alternating current, it will rotate back and forth along the central axis of the torsion beams. Therefore, the scanning angle of the ISM is equal to the sum of the upward deflection angle and downward deflection angle of the micromirror. Due to the symmetrical structure of the micromirror, the angle of upward deflection and downward deflection of the micromirror are theoretically the same, that is, the size of the scanning angle is twice the size of the deflection angle. The data given in table 4 is the deflection angle, and the maximum deflection angle can reach 7.08°, so the Maximum scanning angle of the ISM can reach 14.16°. This scanning angle size is pretty good in the micromirror developed at present. The problem of scanning angle conversion has been explained in the fourth paragraph of section 4.
  10. According to the modification suggestions, we have supplemented the equipment characteristics in paragraph 1 of section 4, introduced the manufacturing process of ISM in paragraph 2 of section 4, improved the test principle in Section 4, and repeatedly checked and modified the full text.
  11. As answered in question 9 above, the scanning angle of the ISM we developed has been as high as 14.16 °, which has a good performance in the field of micromirror research and development. The main purpose of this paper is to improve the mechanical performance of the ISM by optimizing the driving system, so that the electromagnetic ISM can be applied to a wider range of application fields. According to the modification opinions, we have rearranged the main innovations of the paper in the summary and summary.

Reviewer 4 Report

The authors report on the effect of driving system parameters on the performance of a MEMS acting as scanning micromirror. Simulation results and experimental data are compared to each other. The novelty of the configuration is rather low. The topic is relevant, however the quality of the language of the paper is not satisfying in the present form. Therefore, a major revision is required.

Further comments:  

(1) The headline should not start with the redundant phrase “Research on the...”

(2) The term “working characteristics” has to be clearly defined in the beginning, i.e. the key parameters of the system have to be addressed.

(3) The scale in Fig. 2 seems to be too small.

(4) The reader does not know where the fitting factors in Eq. (2) come from, and if the accuracy (3 digits after the  comma) is motivated somehow? The sensitivity of the results of the calculations against a variation of the parameters is also an important information, in particular for the evaluation of the stability and robustness.

(5) The unit “Vpp” should be explained as “peak-to-peak voltage” where it appears for the first time.

(6) Conclusions: “To research the effect” should be replaced by “To study the effect”.

(7) The language in general is too minimalistic, often “the” or “a” are missing, etc.

(8) The novelty of the work has to be clearly addressed.

Author Response

According to the review comments, the revised replies are as follows:

  1. We have deleted the phrase " Research on the..." and changed the headline to "Improvement and optimization of electromagnetic integrated scanning micromirror ".
  2. According to the review comments, we have reorganized this aspect in the full text.
  3. The annotations in Fig. 2 have been properly enlarged.
  4. According to the review comments, we have combed and modified this part in detail. For details, please refer to paragraph 3 of section 3 of the article.
  5. As suggested, we have added explanatory notes where Vpp first appeared.
  6. As suggested, we have changed "To research the effect " to " To study the effect ".
  7. According to the suggestion, we have checked the full text repeatedly and rearranged the inappropriate sentences in the full text.
  8. According to the review comments, we have reorganized the conclusions and clearly stated the innovation points of the paper.

Round 2

Reviewer 1 Report

I have no further comments to add.

Author Response

Thanks for your valuable modification suggestions and your recognition of our work! Sincerely!

Reviewer 2 Report

The authors have taken account of most of the remarks and questions in this revised version of the paper. Several points have been precised and the developed technology may desserves interest in laser scanning applications. The manuscript can now be considered for publication in the journal.

Author Response

(The authors gave the same response as above.)

Reviewer 3 Report

Thanks for the updated paper. Reads better and is more understandable but i feel there is still further work that could be done to enhance this paper without just referencing other papers. 

Although you are focusing on mirror deflection and bias voltage in this paper, you should include abbreviated information to the reader as most are not going to read all the references unless they have true interest in the processes used. A brief fabrication review and structural design concept would be helpful and other short details would be extremely beneficial to the reader. 

Most of your images are the same size, fig 3 is helpful but your Ansys model images are way to small as one can barely read these images. 

I believe you should take an in depth look at other efforts out in the field as scanning micromirrors are very common in literature. Should compare your technique to other actuation methods such as electrothermal, electrostatic, and piezoelectric to name a few and why yours is better (for example, low voltages, scanning angle range, etc.)

Overall, this is a better paper but i think it still needs some work to make it an independent body of work and for completeness. 

Author Response

Thanks for your valuable modification suggestions and your recognition of our work! Based on the modification suggestions, we have made necessary modifications and supplements. The details are as follows:

  1. According to the suggestion for modification, we have added fabrication and structural design in the paper. For details, please refer to the first paragraph of Section 4 of the paper.
  2. We have adjusted the pictures to the most appropriate proportions.
  3. Based on the proposed modification suggestion, we have extensively reviewed relevant literature and added comparisons with other devices in the paper. See section 4, paragraph 5 of the paper for details.

Reviewer 4 Report

The improved version of the manuscript can be accepted for publication now.

Author Response

(The authors gave the same response as above.)

Round 3

Reviewer 3 Report

I believe the authors have satisfactorily responded to my questions and the paper is OK. Thanks for making the modifications as it now is far more technically complete.